# Targeting the Hippo Pathway for Breast Cancer Therapy

**DOI:** 10.3390/cancers10110422

**Published:** 2018-11-05

**Authors:** Liqing Wu, Xiaolong Yang

**Affiliations:** Department of Pathology and Molecular Medicine, Queen’s University, Kingston, ON K7L 3N6, Canada; liqingwu567@gmail.com

**Keywords:** hippo pathway, breast cancer, targeted therapy, YAP, TAZ, LATS, MST

## Abstract

Breast cancer (BC) is one of the most prominent diseases in the world, and the treatments for BC have many limitations, such as resistance and a lack of reliable biomarkers. Currently the Hippo pathway is emerging as a tumor suppressor pathway with its four core components that regulate downstream transcriptional targets. In this review, we introduce the present targeted therapies of BC, and then discuss the roles of the Hippo pathway in BC. Finally, we summarize the evidence of the small molecule inhibitors that target the Hippo pathway, and then discuss the possibilities and future direction of the Hippo-targeted drugs for BC therapy.

## 1. Introduction

### 1.1. Targeted Treatment of Breast Cancer

BC is the most frequently diagnosed cancer among females, accounting for 25% of all cancer cases worldwide [1]. Over the past decades, despite substantial efforts made to improve the survival and quality of life, BC remains a deadly threat for patients. For most types of BC, treatment involves surgery, radiation therapy, hormone therapy, chemotherapy, and the latest targeted therapy [2]. Up to the present day, multiple targeted drugs have been approved in the treatment of BC by FDA as illustrated in Table 1, including inhibitors of estrogen receptor (ER), aromatase, cyclin-dependent kinase (CDK) 4/6, mTOR (rapamycin), and poly(ADP-ribose) polymerase (PARP), and epithelial growth factor receptor (EGFR) and human epithelial growth factor receptor 2 (HER2)-targeted agents [3]. Additionally, studies about the antitumor effects of many other inhibitors such as inhibitors of vascular endothelial growth factor (VEGF), farnesyl transferase, and fibroblast growth factor receptor (EGFR) also show a promising future and have caught attentions as well [4].

However, current treatment has its own limitations. First of all, BC subtypes show different responses to systemic therapy, suggesting the treatment should be more specific for each patient [5,6]. By detecting the expression levels of ER and progesterone receptor (PgR), as well as the HER2 and Ki-67, BC were identified into categories with similar clinical implications, including Luminal A, Luminal B, HER2-positive, and triple-negative breast cancer (TNBC) subtypes [7]. Among them, TNBC shows a relatively poor prognosis, while the Luminal and HER2-positive subtypes respond sensitively to endocrine therapy and targeted therapy [8]. Secondly, drug resistance, especially to monotherapy, has limited the curative efficacy, resulting in a low response rate. Thirdly, the lack of reliable indicators for most of the targeted agents makes it a challenge to select doses and predict the prognosis of patients [9]. Therefore, identifying novel breast cancer therapeutic targets, revealing the mechanisms underlying drug resistances, discovering new biomarkers and developing rational combinations of targeted therapy remain urgent and important.

### 1.2. The Hippo Pathway

The Hippo pathway, named after the *Drosophila* Hpo kinase, is a highly conserved signal transduction pathway that plays important roles in organ size control, tissue regeneration, immune response, stem cell function and tumor suppression [25,26,27]. In mammals, the canonical Hippo pathway consists of four core components that function through phosphorylation: mammalian sterile 20-like kinase (MST; homolog of *Drosophila* Hpo), large tumor suppressor (LATS) kinases, scaffold proteins Salvador homolog 1 (SAV1) and Mps One Binder kinase activator protein 1 (MOB1) (Figure 1). In certain conditions such as high cell density, extracellular matrix stiffness and lack of nutrients, the Hippo pathway is activated, with MST and LATS successively phosphorylated with the support of SAV1 and MOB1 [26]. Then, the activated LATS phosphorylates transcriptional co-activator Yes-associated protein (YAP) and its paralog transcriptional coactivator with PDZ-binding motif (TAZ), which prevent TAZ/YAP from entering the nucleus by anchoring them to 14-3-3 protein and/or promoting their degradation in the cytoplasm (Figure 1) [28,29]. This interrupts their interactions with the transcription factor TEA domain (TEAD) family proteins, which subsequently inhibits cell proliferation and oncogenic transformation and induces apoptosis. Conversely, the dysregulation of the Hippo pathway increases the nuclear functions of TAZ/YAP, leading to active gene expression [30,31], such as several growth-promoting factors, including secretory proteins connective tissue growth factor (CTGF) and CYR61 [32,33], AXL receptor tyrosine kinase [34], c-myc and survivin [35].

Aside from TAZ/YAP-TEAD interaction, TAZ/YAP can also regulate transcription mediated by RUNX, SMADs, TP73, NKX2.1, OCT4 and PPARγ. When the Hippo pathway engages in crosstalk such as with Wnt, TGFβ, Notch and PI3K, the functions of TAZ/YAP are further stimulated [30,32]. With increasing studies, many regulators of TAZ/YAP have been identified in addition to the core Hippo pathway components. For example, TAZ/YAP activity can be regulated in a LATS-independent way, by binding to Angiomotin (AMOT) family proteins, ZO-1/2, α-catenin, β-catenin, PTPN14 and Scribble [36]; the receptor tyrosine kinase EphA2 could activate TAZ/YAP through Rho-ROCK signaling [37]. In this era of targeted therapy, the Hippo pathway appears to be a promising target for the treatment of breast cancer. Here, we summarize the current evidence to demonstrate the mechanisms beneath and provide an overview of the current development of Hippo-targeted therapy for breast cancer.

## 2. The Roles of the Hippo Pathway in Breast Cancer

In 1999, St John et al. discovered that mice lacking *Lats1*, a mouse homology of *Drosophila lats*, display pituitary hyperplasia and develop tumors [38]. Later our studies provided evidence that LATS is a tumor suppressor in human cancer cells [39]. Since then many studies support the role of the Hippo pathway as a tumor suppressor pathway in diverse human cancers, including breast cancer [40,41]. In the following text, we will discuss how each component of the core Hippo pathway is involved in the tumorigenesis and metastasis of breast cancer.

### 2.1. YAP and TAZ

High YAP are more common in BC lacking functional adherens junctions [40]. The expression status of YAP is also associated with the molecular subtypes, tumoral and cellular components of BC, and could be a prognostic marker for patents, pointing to an oncogenic role [42]. YAP overexpression enhances multiple processes for tumorigenesis and metastasis in BC cells, including cellular proliferation, transformation, migration, and invasion [43]. Previous studies reported that overexpression of YAP in human non-transformed mammary epithelial cells induces epithelial-to-mesenchymal transition (EMT), suppression of apoptosis, growth factor-independent proliferation, and anchorage-independent growth on soft agar [44].

Besides, the divergent tumor-suppressive roles of YAP have also been recognized [41,45,46,47,48,49,50,51], which could be explained by many reasons. YAP exists in two major isoforms (YAP1 and YAP2) that may have their own transcriptional targets. Therefore, the relative levels of those two isoforms might decide whether YAP is acting as a tumor suppressor or otherwise in BC cells [52,53]. In addition, YAP1 can translocate to the nucleus and associate with tumor suppressor p73, resulting in apoptosis through transcriptional activation of the proapoptotic gene *puma* [46]. Another explanation for the tumor-suppressive role of YAP is that deregulated TAZ/YAP activity in BC cells induces an anti-tumorigenic immune surveillance response, ultimately leading to the eradication of tumor cells so BC cells have to restrain YAP activity consequently [54,55]. Moreover, studies reported that YAP can bind and signal through anti-apoptotic protein (delta)ΔNp63 isoform to protect cancer cells from DNA damage. Therefore, it is possible that only in certain conditions like DNA damage, YAP can selectively induce p73-mediated apoptosis [56,57]. Additionally, more investigations considering different intrinsic subtypes of BC and stem cells should be done to explain the dramatic effects of YAP [41].

TAZ has also been identified as an oncogene that plays a critical role in the migration, invasion, and tumorigenesis of BC cells [58,59]. It is conspicuously overexpressed in human breast cancer tissues from patients in which its expression levels generally correlate with the TNBC diagnosis [60] and patient prognosis [41]. Overexpression of TAZ in low-expressing MCF10A non-tumorigenic mammary cells leads to the acquisition of a spindle-shaped morphology and increases migratory and invasiveness [58], while the depletion of TAZ inhibits cell growth in 184A1 and HCC1937 breast cancer cells [61] and retards the anchorage-independent growth on soft agar and tumorigenesis in nude mice in MCF7 cells [58]. Additionally, TAZ has been implicated in BC-associated metastatic bone disease, partly through its interaction with hypoxia inducible factor-1α [62]. Recent studies show that TAZ is required for sustaining self-renewal, tumor-initiation capacities [63], and metastatic activity [59] in BC stem cells (BCSCs). The connection between TAZ and BCSCs has been correlated with its interaction with established inducers of the cancer stem cell phenotype such as hypoxia-inducible factor 1 (HIF1) and extracellular cues [64,65,66].

YAP/TAZ act as central players of multiple cancer-associated features such as proliferation and cell survival, migration and metastasis [41], and the tumor-initiating functions. All of these functions rely on their interaction with TEAD transcriptional factors. TAZ/YAP-TEAD complexes directly promote the expression of many oncogenic factors that contribute to BC progression [43], such as cysteine-rich angiogenic inducer 61 (CYR61) and connective tissue growth factor (CTGF) (also known as CCN1 and CCN2, respectively) [32,67,68]. TAZ-TEAD can activate BMP4, which will enhance signaling downstream of TAZ, and then promote Smad1/5 intracellular signaling and cell migration [69]. YAP-TEAD also could control receptor for hyaluronan-mediated motility (RHAMM) transcription leading to ERK activation and cancer metastasis by binding to RHAMM promoter at specific sites [70]. Glutamine metabolism is critical to many tumor cells including BC cells, and it could be regulated by TAZ/YAP. Through the increased expression of downstream genes *SLC1A5* and *GLS*, TAZ/YAP could promote glutamine uptake and therefore upregulate the amount of intracellular glutamine [37]. Besides TEADs, TAZ/YAP can bind to other transcriptional factors, such as the krueppel-like factor 5 (KLF5) and transforming growth factor β (TGFβ)-activated SMAD2/3. The overexpression of YAP could upregulate KLF5 protein levels and mRNA expression levels of its downstream target genes including *FGFBP1* and *ITGB2* that promote BC cell proliferation and survival [71]; the interaction between TAZ/YAP and SMAD2/3 regulates novel targets such as NEGR1 and UCA1 that are necessary for tumorigenic activity in metastatic BC cells [68]. YAP function is also required for cancer-associated fibroblasts (CAFs) to promote matrix stiffening, cancer cell invasion and angiogenesis [72].

### 2.2. Other Components

The upstream components of the Hippo pathway (Figure 1) were found to be tumor suppressors in human breast cancer, and their functions are not limited through the inactivation of TAZ/YAP. In human BC, the downregulation of MST has been identified to be a predictable biomarker for prognosis [73]. MST regulates a diverse array of substrates in addition to the core Hippo pathway components such as LATS, SAV1 and MOB1. Studies have shown that MST also target histone H2B, FOXO, GA-binding protein (GABP) and LATS-related kinases Ndr1/Ndr2 [74,75,76,77], pointing to a tumor-suppressive role.

Clinical evidence has shown reduced expression of LATS in human BC, and functional studies show that overexpression of LATS1 can modulate CDC2 kinase activity and induce pro-apoptotic Bax expression, which causes G2/M cell cycle arrest and induction of apoptosis [78]. Additionally, LATS1 can interact with actin, and Zyxin and LIMK1 [79,80], two regulators of actin filament assembly, regulating actin polymerization [81]. Besides, LATS could phosphorylate angiomotin (AMOT) and thus inhibit cell migration in vitro and angiogenesis [82]. More recently, a direct interaction between LATS and ERα signaling was identified, suggesting that in the presence of LATS, ERα was targeted for ubiquitination and Ddb1–cullin4-associated-factor 1 (DCAF1)-dependent proteasomal degradation, which is a novel non-canonical effect of LATS in the regulation of human breast cell fate [83].

Together, these studies showed that the Hippo pathway is involved in the development of human BC through diverse mechanisms, thus could be a therapeutic target of BC. The therapeutic strategy could be divided into two directions: to inhibit the TAZ/YAP-TEAD interaction, or to up-regulate the upstream components and regulators (Figure 1), since TAZ/YAP activity is mainly governed by LATS kinases [84].

## 3. The Roles of the Hippo Pathway in Therapeutic Drug Resistance of Breast Cancer

### 3.1. Resistance to Chemotherapy

Our laboratory for the first time identified TAZ as a novel gene target responsible for drug resistance in BC. Enhanced levels of TAZ render resistance of mammary epithelial cells to chemotherapeutic drug Taxol (paclitaxel) through the downstream activation of CYR61/CTGF promoters, while TAZ knockdown in TAZ-high/drug-resistant MDA-MB231 BC cells turned them sensitive to Taxol [33]. Later we found that TAZ phosphorylation by CDK1 sensitizes BC cells to antitubulin drugs, suggesting a possible novel target for the treatment of antitubulin-resistant cancers [85]. Moreover, TAZ-expressing BC cells and stem cells were reported to be more resistant than control groups to two widely used chemotherapeutic drugs: doxorubicin and paclitaxel [63,86].

Previous study revealed that the expression of YAP could protect BC cells from chemotherapeutic agents Taxol and cisplatin [44]. Recent evidence suggested that YAP resistance to antitubulin drugs is modulated by a Hippo-independent pathway, since antitubulin drugs activate CDK1 and then YAP is phosphorylated on five sites independent of the Hippo pathway [87]. This result suggests YAP and its phosphorylation status to be novel prognostic predictor for antitubulin treatment for BC patients. Besides, low levels of LATS2 mRNA could be a predictor for favorable response to epirubicin plus cyclophosphamide in breast cancers [88]. This role of LATS2 may be explained by the disruption of the checkpoint function at the G1/S phase induced by down-regulation of LATS2.

### 3.2. Resistance to Targeted Therapy

Several targeted drugs have been approved for the treatment of BC, but the effectiveness varies greatly because of innate and acquired resistance. Current study revealed that the expression levels of TAZ could predict the response to trastuzumab and chemotherapy in Luminal B and HER2-positive BC patients [89]. Besides, in HER2-positive BC cells, TAZ/YAP have been discovered to play a role in the resistance to EGFR/HER2 inhibitor lapatinib by improving the matrix rigidity via the mechanotransduction arm of the Hippo pathway. In HER2-positive BC tumor xenograft mouse model, YAP inhibition increases the sensitivity to lapatinib, suggesting targeting matrix stiffness could be an adjuvant strategy for treating drug-resistant patients [90,91].

### 3.3. Resistance to Endocrine Therapy

Over two-thirds of breast cancer patients express ERα and respond to ERα antagonists (e.g., tamoxifen and fulvestrant), or drugs that reduce ER ligand estrogen (e.g., letrozole). However, resistance to endocrine therapy occurs frequently and the prognosis of patients does not meet the expectations [92]. Phosphorylation of ERα is one of the mechanisms associated with resistance to endocrine therapy, and LATS2 has been demonstrated to activate ERα transcription. LATS2 co-localizes with ERα in the nucleus, and thus contributes to the resistance to tamoxifen and other ER antagonists in ER+ breast cancer [93].

## 4. Current Drugs Targeting the Hippo Pathway for Breast Cancer Treatment

### 4.1. MST and LATS Activation

As mentioned before, MST and LATS are crucial kinases in the Hippo pathway, and they are frequently found to be hypermethylated in BC. Although up till the present moment no direct agents to activate MST and LATS has been discovered, but those indirect activators of MST and LATS still have the potentialities to be targeted drugs for breast cancer cells.

ISIS 5132 is an antisense oligonucleotide designed to hybridize to c-Raf mRNA (Table 2) [94], and Raf-1 was recently found to be an upstream regulator of MST2 [95] (Figure 1). By sequestering MST2 into an inactive complex, Raf-1 could inhibit the apoptosis of BC cells. Preclinical data for ISIS 5132 showed anti-tumor effects in breast cancer and other solid xenograft mouse models, but the agent was withdrawn because of the failure in Phase II clinical trials in patients with colorectal, ovarian or prostate cancer [96]. However, the possibility of ISIS 5132 for the treatment of BC shall need further investigation.

Likewise, since F-actin is the inhibitor of MST/LATS, then the negative regulators of F-actin can indirectly activate MST/LATS activity [97]. The marine-derived macrolides latrunculin are known to reversibly bind actin monomers, disrupting their polymerization. Studies have demonstrated the anti-proliferative and anti-invasive effects of latrunculin in BC cell lines (Table 2) [98,99,100]. Besides, Y27632 could indirectly activate MST/LATS through the inhibition of Rho-associated, coiled-coil containing protein kinase (ROCK) (Figure 1) [101], making itself another targeted agent for BC. Furthermore, anti-Rho siRNAs also could inhibit the proliferation and invasiveness of BC cells in vitro and in vivo [102].

Additionally, statins can also activate LATS through Rho inhibition by suppressing HMG-CoA reductase activity (Figure 1; Table 2) [103]. A case control study suggests that the use of statins is associated with a 51% risk reduction of BC after controlling for age, smoking, alcohol use and diabetes [104]. In a new research, phosphatidic acid (PA) was identified to be a key player in the Hippo pathway, mainly by binding to LATS and NF2 (Figure 1). Therefore, inhibitors of PA and its regulator phospholipase D (PLD) could suppress the oncogenic function of YAP. In MDA-MB-231 cells, PLD inhibitor CAY10594 (Table 2) could suppress cell viability and cell migration in vitro, while another PLD inhibitor FIPI could suppress BC xenograft tumor growth [105]. QLT0267, an integrin-linked kinase (ILK) inhibitor, reduces BC cell growth by activating MST [106]. Moreover, QLT0267 can combine with docetaxel to enhance cytotoxicity, reduce phosphorylated AKT (pAKT) levels, alter F-actin architecture and improve treatment outcomes in an orthotopic BC tumor xenograft mouse model [107].

### 4.2. Targeting TAZ/YAP Regulators

The SRC kinase inhibitor dasatinib is able to inhibits YAP1 nuclear localization and stabilization by reducing Yes-mediated YAP1 phosphorylation or by increasing YAP1 phosphorylation mediated by SRC-PI3K-LATS [108]. Since it suggests promising sensitivity in TNBC cell lines, a Phase II trial examined the efficacy and safety of single-agent dasatinib in unselected patients with advanced TNBC [109]. However, the result showed the effect of single-agent dasatinib is quite limited, so future studies shall investigate other therapeutic settings, such as chemotherapy combinations.

Auranofin was originally used to cure rheumatoid arthritis but recent studies have revealed its antitumor effects [110]. In lung and ovarian cancer, a PKC-AMOT-YAP axis was revealed and auranofin thus could inhibit YAP through AMOT by inhibiting PKC (Figure 1; Table 2) [111]. In breast cancer cells, auranofin induces apoptosis. However, the effect of auranofin could also be explained by prolonged elevation of calcium so further experiments are needed [112].

Additionally, studies showed that taxol also can inactivate TAZ/YAP by activating CDK1 [85,87]. In addition, Bromodomain-containing protein 4 (BRD4), a chromatin-binding protein [113], has been reported to be able to regulate YAP/TAZ transcriptional activity. A potent BRD4 inhibitor named BAY1238097 was tested in human TNBC cell lines and showed satisfying antitumor effect, which might offer new perspectives on the treatment of TNBC patients through modulation of YAP/TAZ (Figure 1; Table 2) [114]. Moreover, energy stress induced by metformin can activate AMP kinase, which directly phosphorylates AMOTL1 and consequently promotes YAP activity in a LATS-independent way (Figure 1; Table 2) [115,116]. A short-term clinical trial in patients with breast cancer showed that, the tumor associated antigen CA15-3 significantly decreased after metformin treatment. However, since metformin may affect breast cancer in other ways as systemic changes in insulin metabolism, more trials should be performed in the future [117].

### 4.3. Inhibition of YAP/TAZ-TEAD Interaction

Through a screen using a luciferase reporter for TEAD response element, verteporfin (VP) was identified to be able to disrupt the interaction between YAP and TEADs (Figure 1; Table 2) in vitro and in vivo [57]. Later, similar study confirmed that VP acts as a potential inhibitor of TAZ/YAP-driven signaling and tumorigenicity in BC [118]. A recent research explored the possibility to combine VP and paclitaxel treatment for patients with TNBC, and the result suggested that both agents are capable of eliminating BC cells and do not interfere with each other [119]. There is an ongoing Phase II clinical trial of continuous low-irradiance photodynamic therapy (CLIPT) using VP for cutaneous BC patients and so far, the null hypothesis of RR ≤ 5% has been rejected [120]. However, VP itself may not be a very promising Hippo-targeted drug for BC treatment due to difficulty for large-scale synthesis, low solubility and stability, and Hippo-independent effects [121,122].

Besides VP, the other members of the porphyrin family, such as hematoporphyrin and protoporphyrin IX are both currently identified as disruptors of YAP-TEAD interaction in xenograft mouse models [123], which could be the next candidates for BC treatment. In addition, current in vitro studies found that tankyrase inhibitor XAV939 could suppress YAP-TEAD transcriptional activities by maintaining the stabilization of AMOT (Figure 1; Table 2) [124,125], while another study revealed XAV939 could reduce tumorsphere formation in TNBC model by suppressing Wnt pathway [126]. Thus, the mechanism of how XAV939 affects BC cells still need more investigations.

Previous studies also show that in BC cells thiazovivin, dasatinib, lovastatin, cucurbitacin I, and pazopanib inhibited YAP-TEAD interaction by changing the nuclear localization of YAP (Figure 1). Among these drugs, dasatinib, statins, and pazopanib are approved as clinically used drugs (Table 2). Therefore, more researches shall be done focusing on those agents. It is notable that pazopanib can also induce proteasomal degradation of TAZ/YAP by the ubiquitin-proteasome system [127]. Additionally, cyclic YAP-like peptides have been designed to occupy the interface 3 on TEAD, which disrupts YAP-TEAD interaction and proves its therapeutic potency [128]. These and the previously mentioned ones are listed in Table 2.

## 5. Ongoing Challenges

There is no doubt that the Hippo pathway plays a critical role in cancer development and therefore presents a promising target for the treatment of BC. However, some ongoing challenges remain urgent and unresolved. First, the exact effects of pathway crosstalk and signaling circuitry on therapeutic outcomes are unknown. For example, a relationship between the Hippo pathway and the Wnt pathway has been discussed for a long time, as the Hippo pathway can restrict or activate the Wnt pathway under certain conditions, while CD44, a target of the Wnt pathway, could interact with the Hippo upstream regulator NF2 and therefore activate the Hippo pathway [129,130]. The Hippo pathway is known to engage other pathways such as TGF-β [131], Ras [132], Hedgehog [133], Notch [134] signaling, so the underlying mechanisms require more investigations. Second, it is clear that the Hippo pathway can serve as a therapeutic target for BC patients, but more experiments and clinical trials shall be conducted to get the data about the sensitivity and the response rate of the Hippo-targeted agents, and to identify reliable biomarkers to predict drug responses. Our lab established the first LATS biosensor that could monitor LATS activity in real-time with high sensitivity non-invasively [135]. Since the system could work in vitro and in vivo, it could be used to examine the effects of the potential targeted agents for BC in mice. Third, we shall discover more novel targets and corresponding inhibitors. For example, PP1A phosphatase was identified to be cable to antagonize the function of LATS, and thus regulate the reversible activation of TAZ, suggesting that PP1A phosphatase inhibitor may acts as an antitumor agent [136]. Additionally, no Hippo-targeted agent has been approved to clinical use for breast cancer yet, so to find out their doses and combination strategies, we still have a long way to go.

## 6. Conclusions and Future Directions

The past decade has witnessed the raising time of the Hippo pathway and the era of targeted therapy. From all the studies mentioned, the Hippo pathway represents both opportunities and challenges for the treatment of breast cancer. In the future, it might be urgent and meaningful to develop drugs that directly target Hippo components (e.g., LATS and YAP/TAZ) or dissociate the TAZ/YAP-TEAD interaction. These drugs can be used alone or combined with other therapeutic drugs (e.g., chemotherapy, target therapy, and immunotherapy) for more effective treatment of drug-resistant or/and metastatic BC in the future.

## Figures and Tables

**Figure 1 cancers-10-00422-f001:**
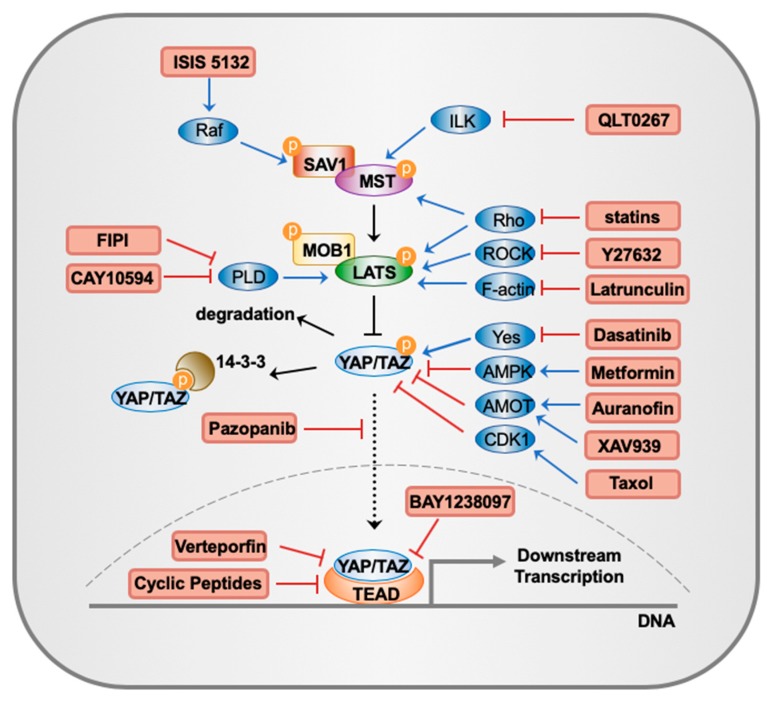
Main components of the Hippo pathway and the current Hippo-targeted inhibitors discussed in this review. In mammals, the canonical Hippo pathway consists of four core components that function through phosphorylation: MST, SAV1, LATS, MOB1. Activated LATS phosphorylates YAP/TAZ, preventing them from entering the nucleus by anchoring them to 14-3-3 protein and/or promoting their degradation in the cytoplasm. This interrupts their interactions with the transcription factor TEAD family proteins, which subsequently inhibits cell proliferation and oncogenic transformation and induces apoptosis. Besides, current Hippo-targeted inhibitors discussed in this review, as well as their targets and major mechanisms, are shown in the figure.

**Table 1 cancers-10-00422-t001:** Approved targeted drugs for breast cancer.

Target	Drugs	Mechanism	References
ER	Tamoxifen	Competitively inhibits the binding of estradiol to ER, resulting in a reduction in DNA synthesis and cellular response to estrogen	[10]
	Fulvestrant	Binds competitively to ER, resulting in ER deformation and decreased estrogen binding	[11]
	Toremifene	Chemically related to tamoxifen, binds competitively to ER	[12]
Aromatase	Anastrozole	Selectively binds to and reversibly inhibits the enzyme aromatase, which catalyzes the final step in estrogen biosynthesis and may result in growth inhibition of estrogen-dependent breast cancer cells	[13]
	Exemestane	Binds irreversibly to and inhibits aromatase	[14]
	Letrozole	Selectively and reversibly inhibits aromatase	[15]
HER2	Trastuzumab	Binds to HER2 on the tumor cell surface, induces an antibody-dependent cell-mediated cytotoxicity against tumor cells that overexpress HER2	[16]
	Pertuzumab	Binds to the dimerization domain of the HER2, therefore prevents the activation of HER signaling pathways, resulting in tumor cell apoptosis	[9]
	Ado-trastuzu-mab emtansine	The maytansinoid DM conjugated to the HER2-targeting transtuzumab is released and binds to tubulin, thereby inhibiting cell division and the proliferation of cancer cells that overexpress HER2	[17]
EGFR, HER2	Lapatinib	Selectively inhibits both EGFR and HER2 tyrosine kinases	[18]
	Neratinib maleate	Binds to and inhibits both HER2 and EGFR	[19]
mTOR	Everolimus	Binds to the immunophilin FKBP-12 to generate an immunosuppressive complex that binds to and inhibits the activation of the mammalian Target of Rapamycin (mTOR)	[20]
CDK4/6	Palbociclib	Selectively inhibits CDK4 and CDK6, thereby inhibiting Rb protein phosphorylation, which suppresses DNA replication and decreases tumor cell proliferation	[21]
	Ribociclib	Specifically inhibits CDK4/6	[22]
	Abemaciclib	Specifically inhibits CDK4/6	[23]
PARP	Olaparib	Selectively binds to and inhibits PARP and PARP-mediated repair of single strand DNA breaks	[24]

ER, Estrogen receptor; HER2, Human epidermal growth factor receptor 2; EGFR, epithelial growth factor receptor; FKBP-12, FK Binding Protein-12; CDK4/6, Cyclin-dependent kinase 4 and 6; Rb, retinoblastoma.

**Table 2 cancers-10-00422-t002:** Agents targeting the Hippo pathway in breast cancer.

Target	Drugs	Major Mechanisms	References
Raf	ISIS 5132	Hybridizes to c-Raf mRNA, stopping the inactivation of MST	[94,95,96]
F-actin	Latrunculin	Activates LATS through regulating F-actin polymerization	[98,99,100]
ROCK	Y27632	Activates LATS through inhibition of ROCK	[101,102]
HMG-CoA	Statins	Activates MST/LATS activity through Rho GTPases	[103,104]
PLD	CAY10594, FIPI	Reduces the production of PA, which could directly bind to and disrupt LATS and NF2	[105]
ILK	QLT0267	Activates MST by inhibiting ILK	[106,107]
Yes	Dasatinib	Activates kinase activity of Yes to activate YAP	[108,109]
PKC	Auranofin	Inhibits YAP through AMOT by inhibiting PKC	[110,111,112]
CDK1	Taxol	Inhibits TAZ/YAP activity by activating CDK1	[87,113]
BRD4	BAY1238097	Interacts with TAZ/YAP and downregulates their transcriptional activities by inhibiting BRD4	[114]
AMPK	Metformin	Inhibits YAP activity by activating AMPK	[115,116,117]
YAP	Verteporfin	Disrupts YAP-TEAD interaction	[57,118,119,120]
AMOT	XAV939	Suppresses YAP-TEAD transcriptional activities by maintaining the stabilization of AMOT	[124,125]
VEGFR & PDGFR	Pazopanib	Inhibits TAZ/YAP nuclear localization by inhibiting VEGFR and PDGFR	[127]
YAP	Cyclic Peptides	Peptides disrupting YAP-TEAD interaction	[128]

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
