# Peer review of "Targeting the Hippo Pathway for Breast Cancer Therapy"

_cancers, 2018, doi:10.3390/cancers10110422_

Round 1

Reviewer 1 Report

In the manuscript entitled “Targeting the Hippo pathway for breast cancer therapy”, the authors systematically summarized the progress of Hippo signaling study in breast cancer development and treatment. This is a well-organized review article. The graph figure/tables are sufficient to illustrate the major points of the text, which are very helpful and efficient in conveying enormous scientific knowledge to the readers.

Minor comments:

1) In Figure 1, XAV939 target is AMOT instead of YAP/TAZ-TEAD complex.

2) Several recently discovered compounds for the Hippo-YAP/TAZ pathway should also be included to make this review article timely for the readers, such as HDAC inhibitors, BRD4 inhibitors, CDK inhibitors, PLD inhibitors, et al.

Author Response

Dear Reviewer,

Thank you for providing us with your thoughtful comments on our manuscript entitled Targeting the Hippo pathway for breast cancer therapy. We have revised our manuscript to conform with your suggestions. Please see our responses to your comments below.

Best regards,

Dr. Xiaolong Yang

Reviewer#1

In the manuscript entitled “Targeting the Hippo pathway for breast cancer therapy”, the authors systematically summarized the progress of Hippo signaling study in breast cancer development and treatment. This is a well-organized review article. The graph figure/tables are sufficient to illustrate the major points of the text, which are very helpful and efficient in conveying enormous scientific knowledge to the readers.”

Minor comments:

1) In Figure 1, XAV939 target is AMOT instead of YAP/TAZ-TEAD complex.
2) Several recently discovered compounds for the Hippo-YAP/TAZ pathway should also be included to make this review article timely for the readers, such as HDAC inhibitors, BRD4 inhibitors, CDK inhibitors, PLD inhibitors, et al.

Response: Thanks for your comments and suggestions. We have revised Figure 1 to make the target of XAV939 more accurate. Besides, we have updated the review by including the most recently discovered compounds such as BRD4 inhibitors and PLD inhibitor, etc. These drugs were also listed and shown in the Table and Figure.

Reviewer 2 Report

In this review article, Wu et. al., discuss the prospects of targeting the Hippo pathway components as an effective strategy to combat Breast carcinoma. In general, the authors cover most parts of present-day knowledge of Hippo components function in Breast cancers and possible inhibition methods applied. The manuscript is well suited for publication and should represent useful information for the scientific community.

There are only a few minor comments that may be incorporated as follows:

1) The role of EphA2 and glutamine metabolism governed by YAP and TAZ in breast cancer. (Sci Signal. 2017;10(508).

2) The contribution of matrix stiffness mediated by TAZ and YAP leading to chemoresistance may be discussed.

Author Response

Dear Reviewer,

Thank you for providing us with your thoughtful comments on our manuscript entitled Targeting the Hippo pathway for breast cancer therapy. We have revised our manuscript to conform with your suggestions. Please see our responses to your comments below.

Best regards,

Dr. Xiaolong Yang

Reviewer#2

In this review article, Wu et. al., discuss the prospects of targeting the Hippo pathway components as an effective strategy to combat Breast carcinoma. In general, the authors cover most parts of present-day knowledge of Hippo components function in Breast cancers and possible inhibition methods applied. The manuscript is well suited for publication and should represent useful information for the scientific community.

There are only a few minor comments that may be incorporated as follows:

1) The role of EphA2 and glutamine metabolism governed by YAP and TAZ in breast cancer. (Sci Signal. 2017;10(508).

2) The contribution of matrix stiffness mediated by TAZ and YAP leading to chemoresistance may be discussed.

Response: Thanks for your comments and suggestions. We added some lines to talk about the role of EphA2 in the introduction of the Hippo pathway, and then we included two paragraphs to discuss how TAZ/YAP regulate glutamine metabolism, and the contribution of matrix stiffness mediated by TAZ/YAP to the drug-resistance in BC, respectively. 

Reviewer 3 Report

A nice addition to the Hippo pathway reviews.

Some typo/grammar corrections:

Line73, evidence

L77, lats to warts/wts

L79, LATS is

L86, patients

L118, novel target

L213, affect 

Author Response

Dear Reviewer,

Thank you for providing us with your thoughtful comments on our manuscript entitled Targeting the Hippo pathway for breast cancer therapy. We have revised our manuscript to conform with your suggestions. Please see our responses to your comments below.

Best regards,

Dr. Xiaolong Yang

Reviewer#3

A nice addition to the Hippo pathway reviews.

Some typo/grammar corrections:

Line73, evidence

L77, lats to warts/wts

L79, LATS is

L86, patients

L118, novel target

L213, affect 

Response: Thanks for your comments and suggestions! We have revised all those typos and grammar mistakes you kindly mentioned. We checked the review again so it shall contain no such mistakes now.

Reviewer 4 Report

Wu et al. present a review describing the Hippo pathway in the context of breast cancer therapy. They discuss current targeted therapy options for breast cancer patients, introduce the Hippo pathway and the roles of the core pathway components in the context of breast cancer studies.  

There are some important omissions in the content. One key gap is the lack of any mention of the controversies that exist on the oncogenic role of YAP. On the one hand, it appears prominent in some tumors, such as KRAS-driven colon, lung and pancreatic cancer, where YAP compensates for loss of oncogenic KRAS. On the other hand, it has a tumor-suppressive role owing to its interaction with the p53 family member p73. This is discussed in the ref 21 cited but not discussed in this review. E.g. Yuan et al. report loss of YAP in breast cancer (Yuan et al., 2008, Cell Death Differ. 15(11): 1752-1759).

A general issue with the clinical translational potential of the Hippo pathway is that many of the studies carried out are on cancer cell lines. There are far fewer studies on clinical cancer cases and the heterogeneity in breast cancer due to the known subtypes adds to the complexity. In this review, there is insufficient distinction between in vitro studies using cancer cell lines, in vivo studies using murine models of breast cancer and clinical studies using tumor material from breast cancer patients. The subtype of breast cancer may be crucial to the context under discussion, but this information is lacking in the text. There should be more critical appraisal of the literature cited in this review in the above contexts.

Other comments:

Line 56: presumably it should read “c-myc and survivin [16].”

Line 58: “current Hippo-targeted agents in breast cancer” is a misleading title since the diagram shows chemical inhibitors of the pathway and not therapeutic interventions for breast cancer patients. The title should be amended.

Line 77: the Drosophila homolog is Wts.

Line 121: Breast cancer subtype and heterogeneity within CAFs may also be important factors here (Costa et al., 2018; Cancer Cell. 33(3): 463-479).

Line 122: this section omits other upstream components such as PNPN14 and KIBRA (Knight et al., 2018, Cell Rep. 22(12): 3191-3205; Moleirinho et al., 2013, Oncogene  32(14): 1821-30).

Line 167/8: “prognosis of patients does meet the expectation” more description of what is the ‘expectation’ should be included to clarify the sentence.

There are some grammatical errors in the text to be corrected.

Author Response

Dear Reviewer,

Thank you for providing us with your thoughtful comments on our manuscript entitled Targeting the Hippo pathway for breast cancer therapy. We have revised our manuscript to conform with your suggestions. Please see our responses to your comments below.

Best regards,

Dr. Xiaolong Yang

Reviewer#4

Wu et al. present a review describing the Hippo pathway in the context of breast cancer therapy. They discuss current targeted therapy options for breast cancer patients, introduce the Hippo pathway and the roles of the core pathway components in the context of breast cancer studies.  

There are some important omissions in the content. One key gap is the lack of any mention of the controversies that exist on the oncogenic role of YAP. On the one hand, it appears prominent in some tumors, such as KRAS-driven colon, lung and pancreatic cancer, where YAP compensates for loss of oncogenic KRAS. On the other hand, it has a tumor-suppressive role owing to its interaction with the p53 family member p73. This is discussed in the ref 21 cited but not discussed in this review. E.g. Yuan et al. report loss of YAP in breast cancer (Yuan et al., 2008, Cell Death Differ. 15(11): 1752-1759).

Response: Thanks for your comments and suggestions! We have discussed the controversies that exist on oncogenic and tumor suppressor functions of YAP in our revised manuscript.

A general issue with the clinical translational potential of the Hippo pathway is that many of the studies carried out are on cancer cell lines. There are far fewer studies on clinical cancer cases and the heterogeneity in breast cancer due to the known subtypes adds to the complexity. In this review, there is insufficient distinction between in vitro studies using cancer cell lines, in vivo studies using murine models of breast cancer and clinical studies using tumor material from breast cancer patients. The subtype of breast cancer may be crucial to the context under discussion, but this information is lacking in the text. There should be more critical appraisal of the literature cited in this review in the above contexts.

Response: in response to reviewer’s suggestion, we have included some paragraphs to introduce the subtypes of breast cancer and added more details to distinguish between different studies in vitro, in vivo and using clinical breast cancer patient samples.

Other comments:
Line 56: presumably it should read “c-myc and survivin [16].”

Response:  “surviving” has been changed to “survivin”, and we checked the review again to avoid typos like this.

Round 2

Reviewer 4 Report

Revised version addresses the reviewer comments and provides context to the information.